# Structure of SALL4 zinc finger domain reveals link between AT-rich DNA binding and Okihiro syndrome

James A Watson*, Raphaël Pantier*, Uma Jayachandran, Kashyap Chhatbar, Beatrice Alexander-Howden, Valdeko Kruusvee, Michal Prendecki, Adrian Bird, Atlanta G Cook

**Spalt-like 4 (SALL4) maintains vertebrate embryonic stem cell identity and is required for the development of multiple organs, including limbs. Mutations in SALL4 are associated with Okihiro syndrome, and SALL4 is also a known target of thalidomide. SALL4 protein has a distinct preference for AT-rich sequences, recognised by a pair of zinc fingers at the C-terminus. However, unlike many characterised zinc finger proteins, SALL4 shows flexible recognition with many different combinations of AT-rich sequences being targeted. SALL4 interacts with the NuRD corepressor complex which potentially mediates repression of AT-rich genes. We present a crystal structure of SALL4 C-terminal zinc fingers with an AT-rich DNA sequence, which shows that SALL4 uses small hydrophobic and polar side chains to provide flexible recognition in the major groove. Missense mutations reported in patients that lie within the C-terminal zinc fingers reduced overall binding to DNA but not the preference for AT-rich sequences. Furthermore, these mutations altered association of SALL4 with AT-rich genomic sites, providing evidence that these mutations are likely pathogenic.**

## Introduction

Embryonic stem cells (ESCs) balance pluripotency with a development and differentiation program to generate distinct tissues within an organised body plan. Proteins involved in development are typically expressed transiently, at specific embryonic locations, and are absent from adult tissues or restricted to specific tissue progenitor cells. SALL4 is a protein of this type which is expressed both in ESCs and in later lineages during embryogenesis and plays critical roles in the development of various organs (Sweetman & Munsterberg, 2006). It is one of four spalt-like C2H2 zinc finger DNA-binding proteins in mouse and humans. SALL4 deficiency leads to peri-implantation lethality in mice (Sakaki-Yumoto et al, 2006) and

increased neuronal differentiation potential in mouse ESCs (Miller et al, 2016), indicating that SALL4 helps maintain stem cell identity. Heterozygous *SALL4* mutation in mice causes defects in multiple organs including the nervous system, limbs, kidneys, heart, and anorectal tract (Koshiba-Takeuchi et al, 2006; Sakaki-Yumoto et al, 2006). Consistent with the phenotypes of SALL4 haploinsufficiency in mice, patients with Okihiro syndrome, an autosomal dominant disorder caused by mutations in SALL4, also present a range of symptoms including limb defects, eye anomalies (Duane syndrome), vertebral malformations, hearing loss, kidney defects, heart anomalies, and anal stenosis (Al-Baradie et al, 2002; Kohlhase et al, 2002). Some Okihiro syndrome patients have a presentation similar to thalidomide embryopathy (Kohlhase et al, 2003); consistent with this, SALL4 is a cellular target of thalidomide, which facilitates binding of SALL4 to the CLR4$^{CRBN}$ E3 ubiquitin ligase that ubiquitylates SALL4 and leads to its destruction (Donovan et al, 2018; Matyskiela et al, 2018, 2020).

SALL4 contains seven zinc fingers (Znfs) arranged in three clusters (Fig 1A). Sequence comparisons suggest that these are closely related to zinc finger clusters (ZFCs) 1, 2, and 4 of SALL1 and SALL3 (Sweetman & Munsterberg, 2006). Two SALL4 isoforms have been reported: SALL4A, which encompasses all three ZFCs, and SALL4B, which only has ZFC4. In ESCs, SALL4B is sufficient to maintain ESC identity in the absence of SALL4A (Rao et al, 2010). The roles of ZFC1 and ZFC2 are less well understood. However, SALL4A but not SALL4B can interact with the transcription factor PLZF in spermatogonial progenitor cells, suggesting a role for ZFC1 and/or ZFC2 in protein–protein interactions (Hobbs et al, 2012). Furthermore, ZFC1 and ZFC2 mediate SALL4 ubiquitination in the presence of thalidomide, leading to SALL4 degradation (Matyskiela et al, 2018). ZFC1 was reported to bind to 5-hydroxymethylcytosine (Xiong et al, 2016) and AT-rich DNA sequences in vitro (Ru et al, 2022). However, a SALL4 truncation lacking both ZFC1 and ZFC2 shows no defect in genome-wide chromatin binding in ESCs, suggesting that these ZFCs are largely dispensable for DNA binding in vivo (Pantier et al, 2021).

Wellcome Centre for Cell Biology, Max Born Crescent, Edinburgh, UK

Correspondence: atlanta.cook@ed.ac.uk
Valdeko Kruusvee's present address is University of Copenhagen, Copenhagen Plant Science Centre, Plant Biochemistry, Copenhagen, Denmark
Michal Prendecki's present address is R&D Center in Poznań, Medicofarma Biotech S.A., Wielkopolska Center of Advanced Technologies, Poznan, Poland
*James A Watson and Raphaël Pantier contributed equally to this work

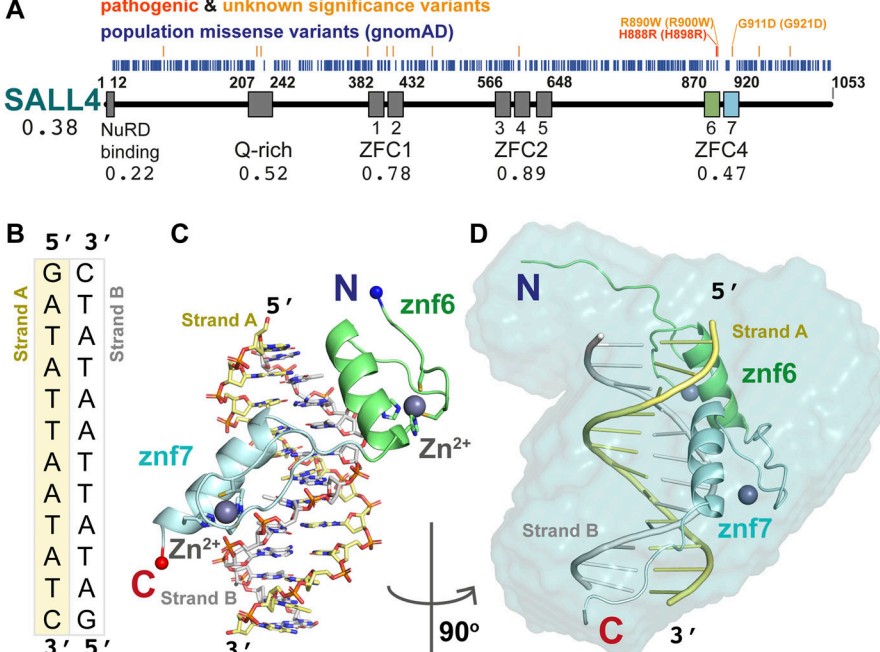

**Figure 1. SALL4 ZFC4 in complex with DNA.**
**(A)** Domain overview of human SALL4 protein. Missense variants from gnomAD population data (blue) are placed relative to the sequence. Above are pathogenic missense mutations (red) and variants of unknown significance (orange) that are absent from the gnomAD database. The NuRD binding region (aa1–12) and Q-rich sequences (aa207–242) are followed by seven zinc finger domains arranged in three clusters (ZFC1, aa382–432; ZFC2, aa566–648; and ZFC4, aa870–920). The variant depletion value for SALL4 (Vp) is given, along with domain-level variant depletion values (VdVp ratios) under the domain labels. A construct of ZFC4 containing zinc fingers 6 (green) and 7 (blue) was used for structural analysis. **(B)** The palindromic DNA sequence co-crystallised with SALL4. Strand A and B are coloured yellow and white, respectively. **(C)** An overview of the mouse SALL4 ZFC4–DNA complex, showing a single protein chain interacting with the DNA. Colour schemes match those of schematics in (A, B). **(D)** An all-atom model of SALL4–ZFC4 complex with DNA fitted into an SAXS envelope. The complex is rotated 90° around the vertical axis relative to (C).
Source data are available for this figure.

We previously showed that SALL1, SALL3, and SALL4, which all encode ZFC4, are all selectively enriched on AT-rich DNA (Pantier et al, 2021). Similar to observations for SALL1 (Yamashita et al, 2007), ZFC4 of SALL4 is required for its localisation to mouse pericentric heterochromatin (Sakaki-Yumoto et al, 2006) and has a strong preference for a range of AT-rich DNA sequence motifs (Kong et al, 2021; Pantier et al, 2021). The molecular basis of this broad specificity is unknown, but there is evidence that it is essential for the ability of SALL4 to maintain stemness in ESCs by sensing differences in sequence composition in the genome (Pantier et al, 2021). Importantly, discrete mutation of ZFC4 leads to precocious ESC differentiation and embryonic lethality, phenocopying complete loss of SALL4 (Sakaki-Yumoto et al, 2006). To gain insight into how SALL4 selects AT-rich sequences, and the likely effect of missense mutations on DNA binding, we undertook a structural, biochemical, and cell-based analysis of ZFC4. We solved the X-ray crystal structure of SALL4 ZFC4 with an AT-rich sequence motif to gain insight into this broad sequence specificity. We also characterised two patient missense mutations that are likely to be deleterious and causative of Okihiro syndrome. We show that these mutations reduce SALL4 ZFC4 binding to AT-rich DNA, yet the proteins retain preference for AT-rich sequences. In cells, full-length mutant proteins fail to localise to heterochromatin. These results confirm that SALL4 binding to AT-rich sequences is fundamental to its in vivo function and that disruptions to this interaction contribute to disease presentation.

## Results

### ZFC4 domain is depleted of population missense variants

According to the gnomAD database (Karczewski et al, 2020), the loss-of-function observed/expected upper bound fraction (LOEUF)

indicates that SALL4 is depleted of inactivating variants and under purifying selection (LOEUF = 0.101). This is consistent with the finding that SALL4 haploinsufficiency is responsible for an autosomal dominant disorder. To further understand the contribution of different SALL4 domains to function, we extracted population missense mutations and calculated an overall missense depletion score for SALL4 protein (Vp) of 0.38 (Deak & Cook, 2022) (Fig 1A). We then considered individual domains of SALL4 and calculated missense depletion relative to the whole protein (VdVp ratio), where a score of ≥1 would indicate that a single domain is not depleted of missense variants compared with the full protein sequence (Deak & Cook, 2022). Three regions were observed to be comparatively depleted of population missense mutations: the N-terminal NuRD binding motif (Lauberth & Rauchman, 2006) (VdVp = 0.22); a glutamine-rich (Q-rich) sequence that has been reported to participate in SALL protein homo- and heterodimer formation (VdVp = 0.50) (Sweetman et al, 2003); and ZFC4, which is essential for SALL4 function in mice (VdVp = 0.47) (Pantier et al, 2021) (Fig 1A). Missense depletion of these regions indicates that they are likely to contribute to the essential functions of SALL4. The two other zinc finger domain regions, ZFC1 and ZFC2, are less depleted (VdVp = 0.78 and 0.89, respectively). Indeed, the gnomAD database, which excludes individuals with severe pathological symptoms compared with the general population, reveals mutations in ZFC1 and ZFC2 (residues 382–432 and 566–648, respectively) that alter the cysteine and histidine residues that are essential for zinc finger integrity (C387Y: Znf1; C412S: Znf2; H644D: Znf5; H644L: Znf5). The absence of both these domains in the shorter SALL4B splice variant is also consistent with a specialised role for these ZFCs in SALL4 function.

A number of likely pathogenic mutations have been reported for SALL4 in Okihiro syndrome patients (Kohlhase et al, 2002, 2003; Borozdin et al, 2004a, 2004b; Kohlhase et al, 2005; Diehl et al, 2015).

We searched the literature and the ClinVar database for missense variants affecting SALL4 as these can inform on functional regions within SALL4 (Landrum et al, 2018).

The only variant with clear evidence for pathogenicity is H888R, which is within ZFC4 (Miertus et al, 2006). Of 56 listed variants of "uncertain significance," 45 are present at equivalent positions in gnomAD, which excludes pathogenic mutations, and are therefore unlikely to cause disease. Of the remaining missense mutations, two map to ZFC4: R890W (VCV000850032.2) and G911D, the second of which is absent from ClinVar but was reported as associated with Okihiro syndrome presentation in a complex genetic alteration (Diehl et al, 2015) (Table S1). As would be expected for pathogenic mutations, these three missense variants are found in a region of ZFC4 that is depleted of population variants (Fig 1A).

## SALL4 ZFC4 binds to AT-rich sequences using polar interactions

To gain insight into SALL4 recognition of DNA, a construct of mouse SALL4 ZFC4 (residues 870–940) was co-crystallised in the presence of a palindromic AT-rich DNA sequence (Fig 1B). This sequence was based on a motif ATATT that was most enriched by SALL4 on systematic evolution of ligands by exponential enrichment (SELEX) (Pantier et al, 2021). Long, needle-like crystals were grown that diffracted to 2.76 Å, with high anisotropy in the diffraction pattern and P1 symmetry. After data reduction, a theoretical model of B-form DNA was used to search for a molecular replacement solution, and four molecules of dsDNA were fitted into the asymmetric unit of the crystal. Subsequently, individual zinc fingers were found, using iterative searches with a model based on PRDM9 (Patel et al, 2017), to complete the asymmetric unit with four copies of ZFC4. Although the stoichiometry of the asymmetric unit is 1:1 for ZFC4 to dsDNA, the ZFC4 chains are not evenly distributed among the DNA molecules, with one copy of the dsDNA lacking any associated protein and one dsDNA binding simultaneously to two ZFC4 chains. The structures were completed through iterative model building and refinement and have good stereochemistry and final $R_{work}$/$R_{free}$ values of 24.7% and 25.4%, respectively (Table S2). All DNA bases are visible in the map. For all ZFC4 chains, residues 880–930 were visible, with chain L extending from 878–933. We base our description on this chain (Fig 1C). Root mean square deviation values for Cα superposition of each of the protein chains ranged from 0.57–0.78 Å, indicating a high level of similarity between all four copies in the asymmetric unit (Fig S1A). Comparison of the refined dsDNA structure with ideal B-form DNA showed that ZFC4-bound DNA has a compressed minor groove and a slightly expanded major groove (Fig S1B).

We measured SAXS scattering curves for ZFC4 alone, dsDNA alone, and ZFC4–DNA complexes as they eluted from size exclusion chromatography (Fig S1C and Table S3). Scattering curves and maximum dimensions ($D_{max}$) of ZFC4, DNA, and the complex were highly consistent with models and measurements from the crystal structure (Fig S1D and E). A bead model calculated from real space analysis of the ZFC4–DNA complex was consistent with a primarily 1:1 protein:DNA stoichiometry in solution (Fig 1D). Normalised Kratky analysis of these data shows that the DNA and SALL4–DNA complex samples show a rise and fall of the curve, whereas ZFC4 shows a continual rise. This indicates that ZFC4 is highly dynamic in solution

and becomes ordered on binding to dsDNA (Fig S1F) (Putnam et al, 2007; Rambo & Tainer, 2011).

Overall, the structure of ZFC4 bound to dsDNA resembles that of other C2H2 zinc finger pairs bound to DNA (Wolfe et al, 2000) (Fig 1C). The helix of each zinc finger probes the major groove of the DNA (Figs 1C and D and 2). The orientation of the Znf6 to Znf7 is similar to that of zinc finger pairs in Zif268 (Elrod-Erickson et al, 1996), indicating that SALL4 ZFC4 belongs to a mode I binding orientation (Garton et al, 2015). We use a common numbering scheme for DNA interacting residues where position 1 is the first residue of the helix and position 7 is the first histidine side chain that interacts with the zinc ion (Wolfe et al, 2000) (Fig 2A). Mode I orientations are promoted by interactions between the residue in position 9 of the first zinc finger with residue in position −2 of the second zinc finger. In ZFC4, these residues are R900 (R890 in human) and T918 (T908 in human), respectively (Fig S2A); mode I zinc finger pairs typically have arginine and serine residues at these positions, but many sequence pairs can be accommodated (Garton et al, 2015).

## Small polar side chains allow ZFC4 to recognise AT-rich sequences

SALL4 differs from other zinc finger proteins in that it recognises a wide variety of AT-rich sequences rather than a fixed DNA sequence. Binding affinity is relatively low, in the micromolar range (Kong et al, 2021). Of note, residues that make up the SALL4 binding interface are predominantly small and polar or hydrophobic (Fig 2B). Specificity in zinc finger proteins is typically conferred by interactions of residues at positions +2, +3, and +6 with bases on the DNA strand that runs 3'->5', with additional contributions from the residue in the −1 position, that can interact with bases on the forward strand. Water molecules typically contribute to base recognition, but, given the limited resolution of our structure, we were not able to fit structured water at the interface. In our co-crystal structure of SALL4 and DNA, the −1 and +2 positions of Znf6 (S891 and S893) are <4 Å from N7 and O6 of G1 on strand A, suggesting that these residues may anchor the protein at the beginning of the palindromic sequence through polar interactions (Fig 2B–D). Were the sequence of strand A to start with an adenine base, similar interactions could be made with N6 and N7. The residue at position +3 (A894) does not make direct contact to DNA but allows a close approach of the −2 position (S890), which is 3.9 Å from the methyl group of T11 on strand B (Fig 2B–D). If this base were an adenine, a hydrogen bond could be formed with N7, suggesting sequence flexibility at this site.

A10, the next base along strand B, does not interact with SALL4. However, there is a close contact of SALL4 with its base pair partner T3 on strand A. The Cα atom of G921, which is at the +2 position in Znf7, is 3.5 Å from the methyl group of T3, suggesting that the presence of a small residue is required for the close approach to this base. T9 is the next base on strand B to be directly recognised, interacting with both the +6 residue (I897) of Znf6 and the −1 residue for Znf7 (T919) (Fig 2B, C, and E). T919 forms a hydrogen bond with O4 of T9. This suggests that Znf6 provides a direct readout for at least one thymine base through hydrophobic and polar interactions. The next base A8 is also directly readout by SALL4, via a bidentate hydrogen bond with N922, the +3 position of Znf7. Minor adjustments in position of N922 could allow a hydrogen bond to

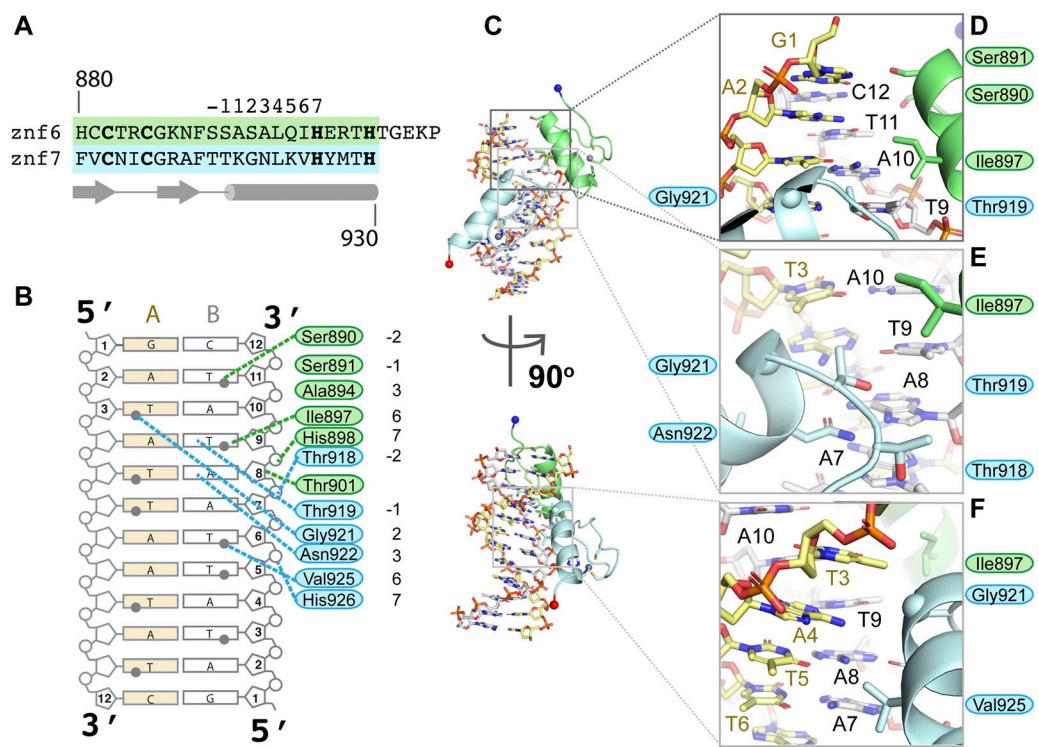

**Figure 2. SALL4 ZFC4 binds DNA with polar and hydrophobic contacts.**
**(A)** Alignment of Znf6 and Znf7 showing standard position annotation for zinc finger helix residues along with secondary structure elements (arrows are beta strands, cylinder is an alpha helix) below. **(B)** Schematic overview of direct interactions between SALL4 ZFC4 and AT-rich DNA. Grey circles represent the methyl groups on thymine bases that point into the major groove. **(C)** Overview of structure showing positions of zoomed views. Lower panel is related to upper panel by a 90° rotation, matching the views in Fig 1C and D. **(D, E, F)** Zoomed in views of side chain interactions with AT-rich DNA.

form with A7, the following base on strand B (Figs 2B, C, and E and S2B). T6 then interacts with small hydrophobic residue V925 at the +6 position of Znf7 (Fig 2B, C, and F). This interaction is similar to the interaction observed between T9 and I897, suggesting that the small hydrophobic side chain provides a good environment for the methyl group of a thymine base. This series of interactions suggest a preference for a core 5'-TAT-3' sequence along the A strand (equating to A10-T9-A8 on the B strand) but that alternative interactions with AT and TA base pairs could be accommodated before, within, and after this core sequence. Recognition of a core 5'-TAT-3' sequence is consistent with previous observations (Kong et al, 2021; Pantier et al, 2021; Ru et al, 2022). The combination of small polar and hydrophobic residues provides an interface where the methyl groups of T bases are accommodated but that can also allow for alternative base interactions with adenine bases.

Previous studies by Garton and colleagues noted that sequence preferences for individual bases are influenced by the relative orientation, or binding mode, of pairs of zinc fingers (Garton et al, 2015). In a large-scale analysis of different possible sequence preferences, they observed that when position 6 is occupied by valine, an A base is typically specified. This fits our observation of V925 interacting with T6, to specify A7 on the forward strand, and I897 interacting with T9, specifying A4 on the forward strand. This study also indicated that when position +2 is occupied by alanine or serine residues, A or A/T preferences are likely to be observed.

SALL4 has S893 and G921 at these positions in Znf6 and Znf7, respectively. In contrast, asparagine at position +3 is normally associated with a C base, whereas we see direct interaction of the +3 residue N922 with A7, specifying a T on the complementary strand.

The sequence of SALL4 ZFC4 is conserved across vertebrates, and all residues that interact with DNA are identical across species (Fig S2C). Furthermore, the ZFC4 sequence is highly conserved with equivalent sequences in SALL1 and SALL3 across the same group of organisms and with *Drosophila* Salr (Fig S2C). Only one residue differs between the ZFC4 domains of SALL4 and SALL1, A892 (mouse), which points away from the DNA binding site. This suggests that SALL1 and SALL3 have an identical DNA binding specificity to SALL4 in ZFC4. SALL1 and SALL3 differ from SALL4 in that they both encode a third ZFC, ZFC3. We carried out an analysis of gnomAD variants for SALL1 and SALL3 and calculated VdVp ratios for the domains in these proteins (Fig S2D). The pattern of missense depletion varies within this family, as would be expected from their apparently differing roles in development (Parrish et al, 2004; Sweetman & Munsterberg, 2006; Warren et al, 2007; Yamashita et al, 2007). All three SALL proteins are depleted of population variants in the NuRD binding motif at the N-terminus and in the Q-rich sequence that is required for SALL protein interactions. In SALL1, all ZFCs show some level of depletion with the lowest VdVp ratios for ZFC1 and ZFC3. In contrast, SALL3 shows the lowest VdVp ratio for ZFC4, whereas ZFC3 in this protein shows no evidence of missense depletion (VdVp = 1.29).

### A hydrophobic residue makes a key contribution to AT-rich DNA recognition

In previous work, we showed that a pair of mutations, T919D and N922A, shows a global loss of DNA binding genome wide (Pantier et al, 2021). These two mutations were based on structure predictions. It is evident from the crystal structure that both T919 and N922 contribute to recognition of the core 5′-TAT-3′ sequence (Fig 2B and D). The structure further suggests that I897 and V925 may provide important hydrophobic interactions that promote binding of thymine bases. To test this hypothesis, we generated mutant proteins containing I897S, V925S, and a double mutation (Figs 3A

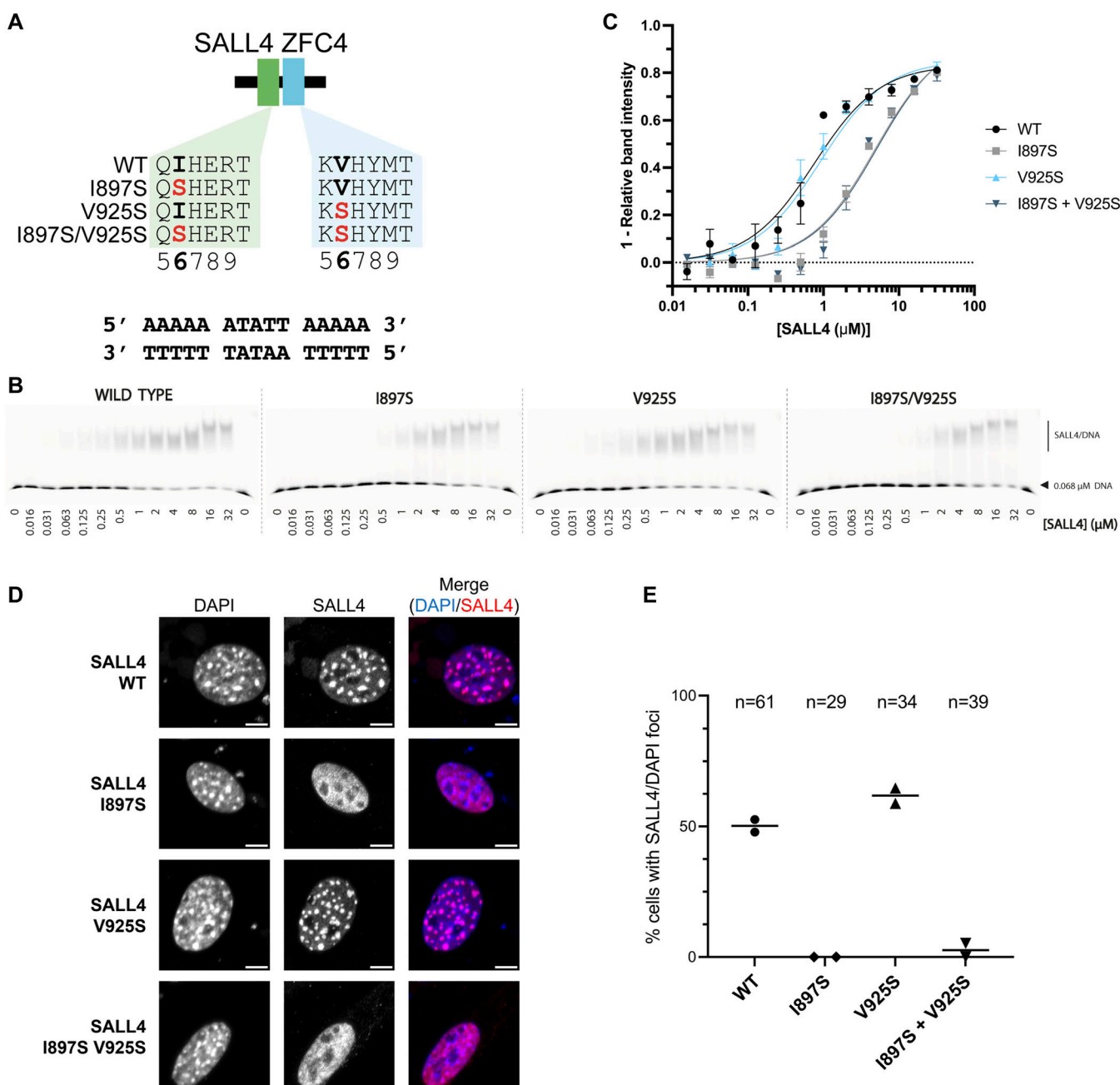

**Figure 3. Mutation of small hydrophobic residues alters SALL4 binding to DNA.**
**(A)** Diagram showing the two residues that were mutated in position 6 of the zinc finger motif. **(B)** EMSA showing WT SALL4 and altered DNA binding interactions of the mutants. Protein concentrations used in titration points are shown below the gel. **(C)** Binding curves for SALL4 WT and mutants based on EMSA data.
**(D)** Immunofluorescence of SALL4 WT and mutant proteins in 3T3 cells transfected with expression constructs, with DAPI staining for comparison. Scale bars are 5 μm.
**(E)** Quantification of cells with SALL4 localisation to DAPI foci. The number of cells analysed in each case is given at the top of the chart. Each data point is from an independent transfection experiment.
Source data are available for this figure.

and S3A and B) and tested binding to DNA in vitro using an electrophoretic mobility shift assay (EMSA) (Fig 3B and C). Compared with WT (apparent dissociation constant [Kd] = 0.76 $\mu$M, Fig 3C and Table S4), single and double mutants (I897S and I897S, V925S) show reduced binding to DNA in vitro (apparent Kd of 4.8 and 5.0 $\mu$M, respectively). Furthermore, these mutations show loss of localisation to heterochromatic foci in cells (Fig 3D and E). The data indicate that mutation of I897 reduces the binding of SALL4 to AT-rich DNA, and this likely results from an alteration of the side chain from a small aliphatic chain to a polar residue. Interestingly, the V925S mutation does not substantially affect SALL4 localisation to pericentric heterochromatin and retains DNA binding affinity (apparent Kd = 0.91 $\mu$M). This is consistent with the observation that small polar side chains likely provide plasticity to the SALL4–DNA binding interface to accommodate different sequences.

### Patient mutations in SALL4 ZFC4 disrupt dsDNA binding in vitro and in cells

Three patient mutations that affect conserved residues of ZFC4 were modelled into the structure to assess their likely impact on SALL4 function (Fig 4A). H888R (mouse equivalent is H898R) is the only established pathogenic missense mutation (Miertus et al, 2006). Although this mutation was proposed to enhance DNA binding, we conclude that this change alters a histidine ligand of the Znf6 zinc ion and so is highly likely to disrupt the fold of Znf6, preventing DNA binding (Fig 4B). R890W (R900W in mouse) is noted in ClinVar (VCV000850032.2), with uncertain significance. Extending from the helix of Znf6, R900 forms bridging interactions with Znf7 via a backbone interaction with T918 (Fig S2A). Furthermore, it forms a closely packed network of interactions with residues of the loop connecting Znf6 and Znf7, including residues E905 and P907. Residues equivalent to R900 (position 9 on Znf6) in other zinc finger proteins play an important role in defining the relative orientation of one zinc finger with respect to the next by interacting with the conserved TGEKP sequence that connects Znf6 and Znf7. Mutation of the TGEKP connector sequence typically affects DNA binding affinity (Wolfe et al, 2000). Mutation of R900 to tryptophan is likely to disrupt the network of close contacts between zinc fingers and could alter the orientation of Znf7 with respect to Znf6 (Fig 4B). Given that the angle between the domains impacts their ability to bind DNA, this mutation is likely to reduce binding to DNA. A third mutation G911D (G921D in mouse) places a larger, negatively-charged side chain at the beginning of the Znf7 helix. G921 mediates close contacts to the major groove (Fig 2F). An aspartate side chain at this position is likely to clash with the DNA bases, potentially altering the overall angle with which Znf7 binds DNA. This mutation is also likely to be pathogenic (Diehl et al, 2015) (Fig 4C).

To test whether uncharacterised patient mutations do indeed alter DNA binding, we purified ZFC4 fragments with mutations R900W and G921D (Figs 4A and S4A and B). Given that H898R is likely to disrupt the fold of the protein, we did not pursue characterisation of this mutation in vitro. EMSA of these proteins showed that both point mutations substantially reduce binding to this probe (apparent Kd = 23 $\mu$M for R900W; a binding constant for G921D could not be determined) (Fig 4D and E and Table S4).

To assess the impact of ZFC4 mutation on DNA binding in cells, full-length mouse SALL4 cDNA carrying the WT sequence or patient missense mutations (H898R, R900W, G921D) was cloned into a mammalian expression vector (Chambers et al, 2003) (Fig S5A). Mouse embryonic fibroblasts (NIH 3T3 cells) were chosen for transfection as they lack expression of endogenous SALL4 and SALL1 and present large nuclear foci with intense DAPI signal, corresponding to AT-rich pericentric heterochromatin (Fig S5B). Strikingly, all mutant proteins showed a diffuse nuclear signal, whereas SALL4 WT co-localised with DAPI bright spots (Fig 4F and G). Localisation or not to DAPI bright spots did not depend on the level of protein expression (Fig S5C). This observation, along with EMSA data (Fig 4D), demonstrates that mutating single residues within ZFC4 is sufficient to disrupt SALL4 binding to AT-rich DNA. Of note, R900W and G921D show similar effects to H898R, indicating that both of these point mutations have an impact on binding equivalent to disrupting the protein fold.

The observations above could either be explained by an overall loss of DNA binding affinity or by a loss of specificity for AT-rich sequences. To investigate whether mutations induced a change in sequence specificity, we performed systematic evolution of ligands by exponential enrichment (SELEX) coupled with high-throughput sequencing (HT-SELEX) (Jolma et al, 2010; Pantier et al, 2021; Pantier et al, 2022) (Fig 5A). ZFC4 WT and mutant (R900W, G921D) proteins were purified and submitted to HT-SELEX, together with a negative control (no protein), to account for PCR bias during the protocol. All possible 6-mer motifs were divided into different categories depending on their proportion of A/T nucleotides. Their relative enrichment was compared across samples at cycle 1, 3, and 6 of HT-SELEX (Fig 5B and Supplemental Data 1). This analysis revealed that ZFC4 WT and both mutant proteins preferentially bind to a large number of AT-rich motifs. However, the level of enrichment was much higher for ZFC4 WT compared with R900W and G921D proteins (Fig 5B). This observation indicates that ZFC4 mutants present decreased DNA binding affinity, in agreement with EMSA data (Fig 4D and E). Most of the enriched DNA motifs by HT-SELEX were shared between ZFC4 WT and mutants, indicating conserved sequence specificity (Fig 5C). As expected, the top motifs were exclusively composed of A and T nucleotides (Fig S6A). Interestingly, the enrichment of DNA motifs correlated better with the total number of A/T nucleotides within a 6 bp motif rather than the number of consecutive A/T nucleotides (Fig S6B). This indicates that A/T base composition is a critical parameter for DNA binding and that SALL4 ZFC4 can tolerate the presence of a single G or C nucleotide within its binding site. Overall, mutations in ZFC4 (R900W, G921D) dramatically reduced DNA binding without affecting preference for AT-rich motifs.

## Discussion

SALL4 is an unusual example of a zinc finger protein that has an expanded specificity for a range of AT-rich sequences. Our structure of SALL4 with an AT-rich DNA sequence shows that SALL4 ZFC4 makes close contacts to bases in the major groove primarily mediated by small hydrophobic or polar side chains that allow hydrogen

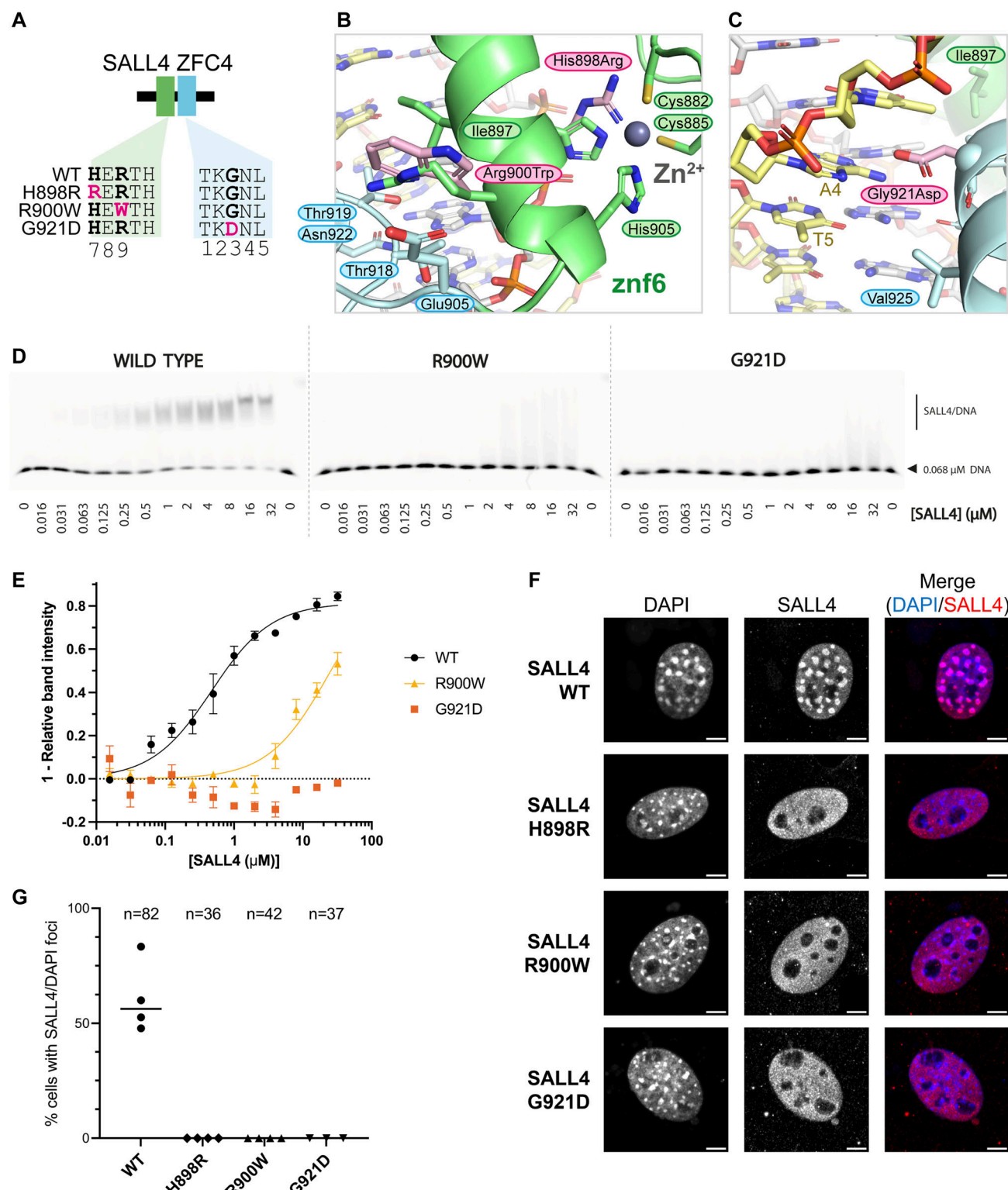

**Figure 4. ZFC4 patient missense mutations reduce SALL4 binding to DNA and alter localisation in nuclei.**
**(A)** Diagram showing SALL4 ZFC4 WT and mutant constructs used in this study. **(B)** Model of point mutations H898R and R900W (mouse numbering, pink carbon atoms) superposed on WT structure to show alterations in proteins structure. H898R would disrupt zinc ion binding. R900W likely disrupts the interface between Znf6 and Znf7. **(C)** Zoomed in view showing the position of the G921D mutation in the major groove. **(D)** EMSAs showing binding interactions of SALL4 ZFC4 WT and mutant proteins with an AT-rich DNA motif. Protein concentrations used in titration points are shown below the gel. **(E)** Binding curves for SALL4 WT and mutants based on EMSA data. (F) Immunofluorescence of SALL4 WT and mutant proteins in 3T3 cells transfected with expression constructs, with DAPI staining for comparison. Scale bars are 5 $\mu$m.

bonding interactions. Two small aliphatic residues I897 and V925 provide hydrophobic surfaces that interact with methyl groups at C5 on thymine that point into the major groove. Mutation of I897 to serine, a small polar residue, reduces binding of SALL4 to an AT-rich motif. However, a similar mutation of V925 shows retention of DNA binding. A double mutation behaves like a single I897S mutation, indicating that this residue has a larger impact on affinity and specificity. Together, the data show that the small size and non-charged nature of DNA-binding residues in SALL4 allow a close association of the zinc fingers to the major groove, allowing recognition of diverse sequences, with a concomitant narrowing of the minor groove.

Previously, we showed that a double point mutant of SALL4 (T919D, N922A) had a cellular phenotype equivalent to deletion of ZFC4 (Pantier et al, 2021). Our structure reveals that these two residues indeed play important roles in DNA recognition, as predicted (Fig 2D). Our SELEX data indicate that more than one G/C base pair is not well tolerated within SALL4 binding sites (Fig S6B). Modelling of GC base pairs onto AT base pairs in the structure shows that the loss of the methyl-5 group, on mutating T to C, changes the major groove surface. Key interactions with the I897 and T919 are lost, and a more polar surface is presented in the major groove. A GC-rich sequence would also add bulk in the minor groove, which would likely make the DNA structure less able to compress in the minor groove (Fig S1B). It is possible that G/C base pairs are selected against because they are more polar than A/T base pairs or that A/T base pairs permit more compression of the minor groove.

Residues that contact DNA in our structure are highly conserved among SALL4 proteins (Fig S2C [Pantier et al, 2021]) with SALL1 and SALL3 showing identical amino acids at positions that bind DNA in ZFC4. This suggests that, at the level of domains, our data give important insights into DNA binding for all three SALL proteins. The highly similar sequences and expression profiles of SALL4 and SALL1 suggest some functional redundancy. Like SALL4, SALL1 protein is expressed in ESCs, is targeted to heterochromatin, and forms homo- and heterodimers with SALL4 (Yamashita et al, 2007; Rao et al, 2010). Indeed, genetic deletion of both *Sall4* and *Sall1* results in stronger phenotypes than either single mutation both in ESCs and mice (Sakaki-Yumoto et al, 2006; Miller et al, 2016). Okihiro syndrome has an overlap in presentation with Townes–Brocks syndrome, which is caused by mutations in the *SALL1* gene (Kohlhase et al, 2002), further indicates that these two proteins have overlapping functions.

Most of the patient mutations described for SALL4 are nonsense or insertion/deletion mutations that are likely to cause loss-of-function of the gene, with consequent haploinsufficiency. The effects of missense mutations are less clear. We noted two uncharacterised patient missense mutations, along with H888R, that map to ZFC4 in regions highly depleted of population missense variants. Our previous work has established that specific disruption of ZFC4 in an otherwise intact SALL4 protein leads to embryonic lethality in mice, demonstrating the importance of this DNA binding domain (Pantier et al, 2021). Our biochemical and cellular characterisation of SALL4 ZFC4 missense mutations showed disrupted DNA binding in vitro and in cells. Although H888R was already linked with Okihiro syndrome (Miertus et al, 2006), our study provides experimental evidence that G911D (Diehl et al, 2015) and R890W (ClinVar, VCV000850032.2) are also likely to be disease-causing mutations.

Our HT-SELEX analysis on 6-mer motifs (based on coverage of the major groove by ZFC4 in the crystal structure) is similar to our previous study on 5-bp motifs (Pantier et al, 2021). SALL4 ZFC4 binds to a wide range of AT-rich DNA motifs, potentially allowing the protein to "read" DNA base composition. Interestingly, although the patient mutations reduce binding to DNA, the proteins still retain AT-rich specificity. In the case of G921D, this is likely to be because only Znf7 is affected by the mutation, and some specificity will be retained from Znf6. In the case of R900W, the prediction is that the orientation between the zinc fingers is likely to be altered. However, each individual zinc finger is still likely to be able to interact with DNA. This suggests that the loss of affinity is likely to be because the two zinc fingers cannot optimally interact with DNA at the same time.

Overall, our structural, biochemical, and cell-based data show that ZFC4 presents a highly conserved binding interface with DNA. The hydrophobic and polar residues that make up this interface likely provide a flexible interface that allows optimal interaction with methyl groups from thymine residues. Patient missense mutations that alter DNA binding have a major impact on SALL4 localisation in cells even though a preference for AT-rich sequences is retained. This suggests that the DNA binding affinity of SALL4 plays an important role in determining protein localisation and transcriptional silencing in cells.

# Materials and Methods

### Primary sequence analysis

Domain boundaries of SALL proteins were identified based on UniProt annotations (Gasteiger et al, 2005) and previous sequence analyses (Pantier et al, 2021). Missense mutations from gnomAD were processed using 1D–3D and VdVp_calculator scripts (Deak & Cook, 2022). For SALL4, missense variants categorised as "pathogenic" or of "uncertain significance" were extracted from ClinVar. To exclude non-pathogenic mutations, these variants were compared with gnomAD variants and equivalent mutations were removed. Sets of variants were plotted using Plot Protein (Turner, 2013).

### SALL4 ZFC4 cloning and purification

Mouse SALL4 (Q8BX22-1) coding sequence (encompassing codons of residues 870–940) was cloned into a pET-based expression vector as a hexahistidine-GST–tagged fusion protein. Point mutants

**(G)** Quantification of cells with SALL4 localisation to DAPI foci. The number of cells analysed in each case is given at the top of the chart. Each data point is from an independent transfection experiment.
Source data are available for this figure.

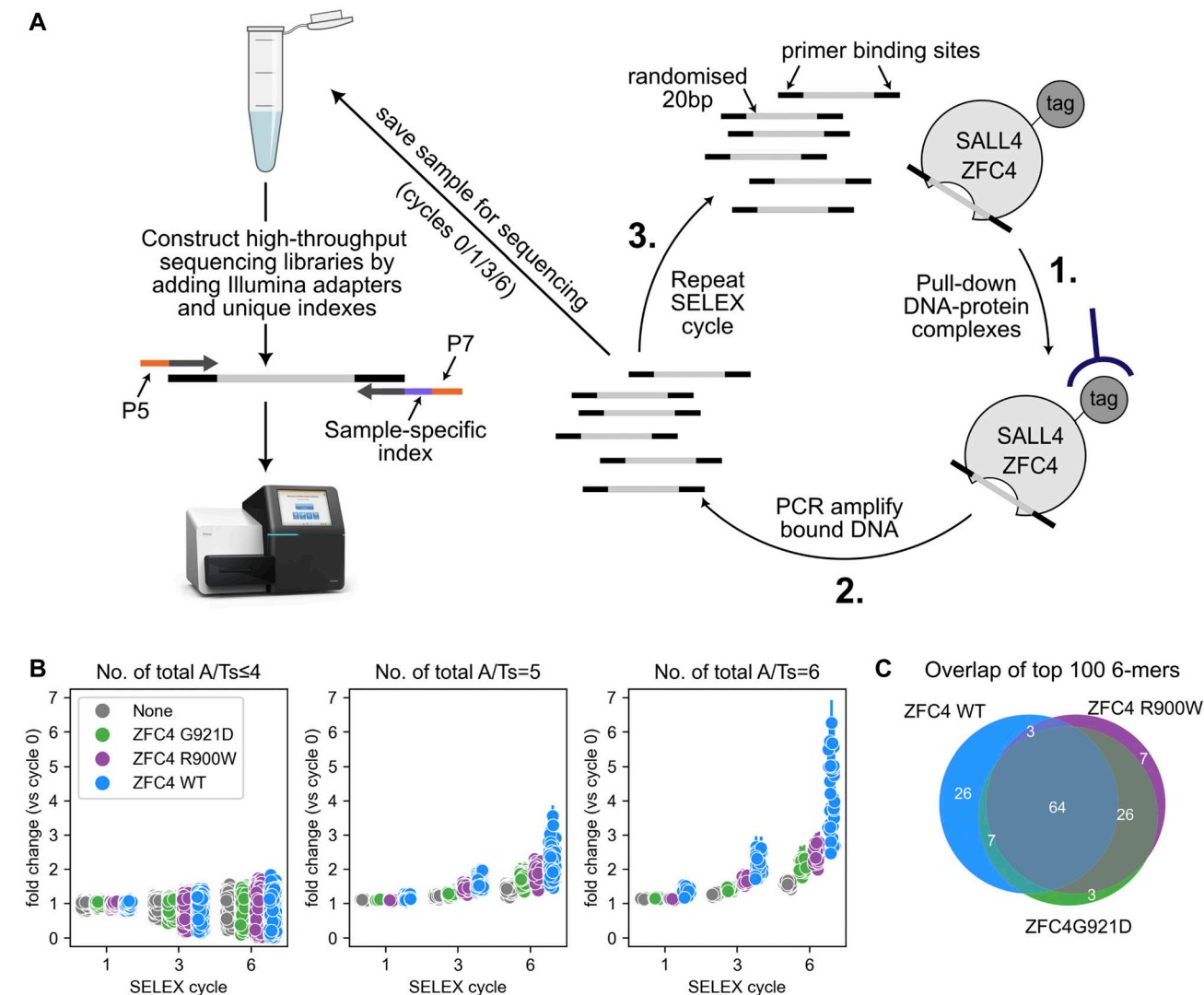

**Figure 5. ZFC4 patient missense mutations do not alter sequence preference.**
**(A)** Diagram summarising the HT-SELEX procedure to determine ZFC4 binding specificity. **(B)** Relative enrichment of 6-mer DNA motifs categorised by total number of A/Ts at cycle 1, 3, and 6 of HT-SELEX with SALL4 ZFC4 WT (blue), R900W (purple), G921D (green), and negative control (grey) samples. Error bars indicate the variability (SD) in three independent replicate experiments. **(C)** Venn diagram showing the overlap of the top 100 enriched 6-mer DNA motifs at cycle 6 of HT-SELEX with SALL4 ZFC4 WT and mutant proteins.
Source data are available for this figure.

were introduced using whole plasmid amplification with Pfu Ultra II (600670-61; Agilent Technologies) and complementary primers, followed by DpnI digestion, transformation, plasmid preparation, and sequencing. These constructs were expressed in BL21 (*DE3*) cells and induced overnight at 20°C with 1 mM IPTG. Cells were lysed using a cell disruptor (Constant Systems) in a buffer containing 20 mM Tris–HCl, pH 7.5; 200 mM NaCl; 0.5 mM $\beta$-mercaptoethanol with protease inhibitor cocktail (Roche) and DNase I (Sigma-Aldrich). The clarified lysate was allowed to bind in batches to GSH resin (Cytiva) and eluted using lysis buffer containing 20 mM reduced glutathione. The GST tag from the eluted proteins was cleaved using rhinovirus 3C protease during dialysis (20 mM Tris–HCl, pH 7.5; 50 mM NaCl; 0.5 mM $\beta$-mercaptoethanol). The cleaved proteins were then purified on a 6-ml Resource S (Cytiva) ion exchange column, and the proteins were

eluted using a salt gradient ranging from 50–1,000 mM NaCl. The eluted proteins were then further purified by size exclusion chromatography (Superdex S75; Cytiva) in 20 mM Tris–HCl, pH 7.5; 200 mM NaCl.

For expression in mammalian cells, mouse SALL4 coding sequence was subcloned into pPYCAG expression plasmids carrying a constitutive CAG promoter (Chambers et al, 2003) (Fig S5A). Equivalent ZFC4 patient mutations were introduced by subcloning mutations from expression plasmids and incorporation using Gibson assembly (NEBuilder HiFi E2621S; NEB). Plasmids are available upon request.

### Crystallisation and structure solution

An equimolar mixture of SALL4 protein with palindromic oligonucleotide (5′-GATATTAATATC-3′) was set up (18 nmol + 18 nmol),

**Life Science Alliance**

giving a final protein concentration of 1.9 mg/ml. The complex was crystallised in 50 mM MES, pH 6.0; 20% PEG 3350; 60 mM $MgCl_2$. Cryoprotectant solution was made by supplementing well buffer with 30% glycerol and added to the drops before harvesting and flash cooling crystals in liquid nitrogen. Data were collected at Diamond Light Source beamline i04. Data were reduced using AUTOPROC with anisotropy correction done by STARANISO (Vonrhein et al, 2011; Tickle et al, 2018). Molecular replacement was carried out using calculated models of B-form DNA (COOT [Emsley & Cowtan, 2004; Emsley et al, 2010]) in PHASER (McCoy et al, 2007), followed by a search model based on PDB ID 5v3g (Patel et al, 2017) and prepared using CHAINSAW (Stein, 2008). The structure was refined in PHENIX with rebuilding in COOT (Emsley & Cowtan, 2004; Emsley et al, 2010). Validation was carried out using MolProbity (Chen et al, 2010) and figures were generated using PyMOL (Schrodinger, LLC, 2015).

### Electrophoretic mobility shift assay

A concentration series of purified untagged SALL4 WT, and mutant proteins were incubated with 68.1 nM DY681 labelled dsDNA in assay buffer (20 mM HEPES, pH 7.5; 150 mM potassium acetate; 5 mM magnesium acetate; and 10 ng/μl poly[deoxyinosinic-deoxycytidylic] acid sodium salt [Sigma-Aldrich]). A total reaction volume of 12 μl was incubated on ice for 30 min, after which 3 μl of native loading buffer (40% sucrose, 0.1 mg/ml BSA, 0.025% bromophenol blue) was added. 10 μl of this reaction was loaded onto a 4% native polyacrylamide gel and separated at 100 V, 4°C, in 0.5×TBE buffer. After an hour, the gel was imaged using a Bio-Rad ChemiDoc MP imaging system set to a 715/30 emission filter.

Unbound DNA bands were quantified using Bio-Rad ImageLab and converted to "1 − relative band intensity" using

$$Y = 1 - \left[ \frac{D_x - D_{bound}}{D_0 - D_{bound}} \right]$$

where $D_x$ is the unbound DNA band intensity at a given SALL4 concentration X, $D_0$ is the unbound DNA band intensity at 0 μM SALL4, and $D_{bound}$ is the quantification of an area equal to a DNA band but in an empty lane (comparable to 100% DNA bound). Data were plotted in Prism 9 (GraphPad) and an isotherm fitted using

$$Y = \frac{Bmax * X}{Kd + X}$$

where Bmax is the maximum fraction bound and Kd is the dissociation constant.

### SAXS

SEC-SAXS experiments were performed at Diamond Light Source on the B21 beamline. Samples at 5–7 mg/ml were injected onto a Superdex S200 Increase 3.2/300 size exclusion chromatography column in 20 mM Tris, pH 7.5; 200 mM NaCl at 0.1 ml/min. SAXS data were recorded using a 3 s exposure. The ATSAS 3.0.5 suite of software was used for processing data (Manalastas-Cantos et al, 2021). CHROMIXS was used for frame selection and sample–solvent subtraction

(Panjkovich & Svergun, 2018). Guinier and distance distribution analyses were carried out using PRIMUS (Konarev et al, 2003). Ab initio bead models were generated with DAMMIF launched from within PRIMUS (Franke & Svergun, 2009). 15 Å density maps were generated from each bead model and the corresponding crystal structures docked into this density using ChimeraX (Pettersen et al, 2021). Additional residues were modelled onto the crystal structure using COOT to match the whole complex used in the SAXS experiment. These models were also fitted to the experimental SAXS data using CRYSOL (launched from PRIMUS) (Svergun et al, 1995).

### Cell culture

Mouse ESCs (Hooper et al, 1987) were grown in Glasgow Minimum Essential Medium (GMEM; cat. 11710035; Thermo Fisher Scientific) supplemented with 15% FBS (batch tested), 1× L-glutamine (cat. 25030024; Thermo Fisher Scientific), 1× MEM non-essential amino acids (cat. 11140035; Thermo Fisher Scientific), 1 mM sodium pyruvate (cat. 11360039; Thermo Fisher Scientific), 0.1 mM 2-mercaptoethanol (cat. 31350010; Thermo Fisher Scientific), and 100 U/ml leukemia inhibitory factor (LIF, batch tested). NIH 3T3 mouse fibroblasts (ECACC, 93061524) were grown in DMEM (cat. 41966; Thermo Fisher Scientific) supplemented with 10% FBS. All cell lines were incubated in gelatin-coated dishes at 37°C and 5% $CO_2$.

For immunofluorescence, $1.2 \times 10^4$ cells were seeded in gelatinised chambered coverslips (cat. 80286; Ibidi). Cells were transfected with 2 μg of SALL4 expression plasmid (pPYCAG-Sall4 WT/H898R/R900W/G921D) using the Lipofectamine 3000 reagent (cat. L3000008; Thermo Fisher Scientific) and following manufacturer's instructions.

### Immunofluorescence

1 d after transfection, cells were washed with PBS and fixed for 10 min at room temperature with a 4% (wt/vol) paraformaldehyde solution. After fixation, cells were washed with PBS and permeabilised for 10 min at room temperature in PBS supplemented with 0.3% (vol/vol) Triton X-100. Samples were incubated for 2 h at room temperature in blocking buffer: PBS supplemented with 0.1% (vol/vol) Triton X-100, 1% (wt/vol) bovine serum albumin, and 3% (vol/vol) goat serum (cat. G9023; Merck Life Science). After blocking, samples were incubated overnight at 4°C (with gentle mixing) with primary antibodies diluted at the appropriate concentration in blocking buffer (Table S5). After 4× washes in PBS supplemented with 0.1% (vol/vol) Triton X-100, samples were incubated for 2 h at room temperature (in the dark) with secondary antibodies conjugated with Alexa Fluor Plus dyes (cat. A32723 or cat. A32733; Thermo Fisher Scientific) diluted 1:500 in blocking buffer. Cells were washed 4× times with PBS supplemented with 0.1% (vol/vol) Triton X-100. DNA was stained with DAPI for 5 min at room temperature, and cells were washed a final time with PBS. Samples were mounted on coverslips using the ProLong glass mounting medium (cat. P36980; Thermo Fisher Scientific), following manufacturer's instructions. Samples were imaged using the Zeiss LSM 880 microscope with Airyscan using a 100× oil objective. Images were analysed and processed using the software Fiji. For each transfection

experiment, all SALL4-positive cells were counted and categorised according to their nuclear expression pattern (foci or diffuse signal).

### HT-SELEX

SELEX coupled with high-throughput sequencing (HT-SELEX) was performed as previously described (Pantier et al, 2022), in triplicate experiments. Oligonucleotides were ordered from Integrated DNA Technologies. Throughout the protocol, SELEX libraries were amplified by PCR using the high-fidelity Phusion DNA polymerase (cat. M0530L; NEB) and purified using the MinElute PCR Purification Kit (cat. 28004; QIAGEN). Purified, recombinant SALL4 ZFC4 WT, R900W, and G921D (residues 870–940) were used in SELEX reactions. SELEX libraries (1.5 $\mu$g for the first cycle, 200 ng for subsequent cycles) were mixed with 1 $\mu$g of recombinant ZFC4 WT or mutant proteins in 100 $\mu$l of SELEX buffer (50 mM NaCl; 1 mM MgCl$_2$; 0.5 mM EDTA; 10 mM Tris–HCl, pH 7.5; 4% glycerol) freshly supplemented with 5 $\mu$g/ml poly(dI-dC) (cat. P4929; Merck Life Science) and 0.5 mM DTT. A negative control experiment (without addition of proteins) was also performed to control for technical bias during the SELEX protocol. After a 10-min incubation at room temperature, 50 $\mu$l of Ni$^{2+}$ Sepharose 6 Fast Flow Beads (cat. 17531806; Cytiva), previously equilibrated in SELEX buffer, was added to each sample to capture protein-DNA complexes. After a 20-min incubation at room temperature, beads were washed five times with 1 ml of SELEX buffer to remove unbound oligonucleotides. After the final wash, beads were resuspended in 100 $\mu$l H$_2$O and used directly for PCR amplification. For each SELEX sample, optimal PCR conditions were empirically determined by running the same PCR reaction several times with increasing number of cycles. Amplified and purified SELEX libraries were used as input for subsequent rounds of SELEX, up to 6× cycles. To generate samples for high-throughput sequencing, SELEX libraries were amplified using primers containing Illumina adapters and unique indexes. HT-SELEX libraries were pooled in equimolar amounts, and contaminating primers were eliminated by performing a clean-up with KAPA Pure beads (cat. 07983271001; Roche), using a 3× beads-to-sample ratio. The HT-SELEX library pool was submitted to high-throughput sequencing using the Illumina MiSeq platform (EMBL GeneCore facility).

### HT-SELEX analysis

All possible canonical k-mer sequences (k = 6) were searched individually in SELEX libraries at different cycles using eme_selex (Pantier et al, 2022). A canonical sequence of a k-mer pair is the lexicographically smaller of the two reverse complementary sequences. For every k-mer, the number of reads containing the k-mer is normalised by the total number of reads in the library to generate a fraction. To quantify the abundance of the k-mer, fold change of fraction at higher SELEX cycle(s) versus fraction at initial random library (cycle 0) is calculated. This fold change (versus cycle 0) is visualized for k-mers grouped according to the total number of A/Ts and consecutive number of A/Ts. Top 100 abundant canonical k-mers from ZFC4 WT and mutant HT-SELEX experiments at SELEX cycle 6 are used to visualize the overlap using a Venn diagram. Top 9 abundant k-mers from ZFC4 WT SELEX library at cycle 6 are searched allowing one mismatch and a position frequency matrix is generated.

Subsequently, the position frequency matrix is used to visualize the motif logos. Raw and processed HT-SELEX data are deposited in the ArrayExpress database at EMBL-EBI (www.ebi.ac.uk/arrayexpress) under accession number E-MTAB-11519. Source code to reproduce the analysis is available at https://eme-selex.readthedocs.io.

## Supplementary Information

## Acknowledgements

We thank David Kelly and the COIL facility for microscopy support, Vladimir Benes (EMBL GeneCore facility, Germany) for support with high-throughput sequencing. We thank Diamond Light Source i04 and B21 staff for assistance with data collection. Coordinates are deposited in the PDB with code 8A4I; SASDB accession codes are SASDP64—SALL4 ZFC4; SASDP74—DNA; SASDP84—SALL4 ZFC4 + DNA; HT-SELEX data are deposited at Array express E-MTAB-11519. Original data images are available at https://datashare.ed.ac.uk/handle/10283/4773. This work used the Edinburgh Protein Production Facility and the Centre Optical Instrumentation Laboratory funded by Wellcome Core Grants 092076 and 203149 to the Centre for Cell Biology. AG Cook is supported by a Wellcome Senior Fellowship (200898). A Bird holds a Wellcome Investigator Award (107930) and a European Research Council Advanced grant (EC 694295 *Gen-Epix*) and is a member of the Simons Initiative for the Developing Brain, University of Edinburgh.

### Author Contributions

JA Watson: data curation, formal analysis, investigation, visualization, writing—review and editing, found and optimised crystal conditions for the palindromic DNA partner, and carried out primary sequence analyses, SAXS measurements/analysis, and ESMA assays.

R Pantier: conceptualization; formal analysis; investigation; visualization; writing—original draft, review, and editing; performed HT-SELEX on ZFC4 constructs; carried out localisation studies of SALL4 mutants in cells; co-conceived the project; and co-wrote the article.

U Jayachandran: investigation, visualization, writing—review and editing, prepared mutants, and carried out binding assays.

K Chhatbar: formal analysis, investigation, visualization, writing—review and editing, analysed HT-SELEX data, and generated figure panels.

B Alexander-Howden: investigation and writing—review and editing.

V Kruusvee: investigation, writing—review and editing, developed purification protocols for different SALL4 constructs, and found initial crystals for SALL4–DNA complexes.

M Prendecki: investigation, writing—review and editing, and developed initial purification protocols for SALL4/DNA complexes.

A Bird: conceptualization; supervision; funding acquisition; writing—original draft, review, and editing; co-conceived the project; and co-wrote the article.

AG Cook: conceptualization, formal analysis, supervision, funding acquisition, investigation, visualization, and writing—original draft, review, and editing. Solved and refined the structure. Co-conceived the project and co-wrote the article.

**Life Science Alliance**

## Conflict of Interest Statement

The authors declare that they have no conflict of interest.

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
