## [Reviewer comments · Life Science Alliance]

Life Science Alliance

Structure of SALL4 zinc-finger domain reveals link between AT-rich DNA binding and Okhiro syndrome

James Watson, Raphaël Pantier, Uma Jayachandran, Kashyap Chhatbar, Beatrice Alexander-Howden, Valdeko Kruusvee, Michal Prendecki, Adrian Bird, and Atlanta Cook

DOI: <https://doi.org/10.26508/lsa.202201588>

Corresponding author(s): Atlanta Cook, University of Edinburgh

Review Timeline:

Submission Date:	2022-07-04
Editorial Decision:	2022-08-07
Revision Received:	2022-11-28
Editorial Decision:	2022-12-16
Revision Received:	2022-12-19
Accepted:	2022-12-20

Scientific Editor: Novella Guidi

Transaction Report:

August 7, 2022

Re: Life Science Alliance manuscript #LSA-2022-01588-T

Dr. Atlanta G Cook
University of Edinburgh
Wellcome Trust Centre for Cell Biology
Michael Swann Building, King's Buildings
Max Born Crescent
Edinburgh EH9 3F
United Kingdom

Dear Dr. Cook,

Thank you for submitting your manuscript entitled "Structure of SALL4 zinc-finger domain reveals link between AT-rich DNA binding and Okhiro syndrome" to Life Science Alliance. The manuscript was assessed by expert reviewers, whose comments are appended to this letter. We invite you to submit a revised manuscript addressing the Reviewer comments.

Thank you for this interesting contribution to Life Science Alliance. We are looking forward to receiving your revised manuscript.

Sincerely,

B. MANUSCRIPT ORGANIZATION AND FORMATTING:

Reviewer #1 (Comments to the Authors (Required)):

SALL4 is a multi-zinc finger (ZnF) factor that is mutated in Okhiro syndrome; its expression is reactivated in many aggressive cancers where it has been considered as a candidate therapeutic target. However, its mode of interaction with DNA is not well understood. Two recent studies, one from this group of authors (Pantier et al Mol Cell 2021) and another investigating liver cancer (Kong et al Cell Reports 2020) found that the C terminal pair of ZnFs 6 & 7 in SALL4 bind AT rich sequences and are critical for the function of the protein.

In this manuscript, the authors describe the structure of the C-terminal ZnF cluster 4 of Sall4 in contact with an AT rich DNA motif. The paper is a focused study that provides valuable structural information about this important transcriptional regulator, though the results agree with prior analyses of how ZnFs interact with DNA, somewhat reducing the overall significance of this work for the field. In addition, I find that the functional analysis of the SALL4 mutants and the studies investigating the specificity/affinity of the SALL4 binding site are superficial. In my opinion the manuscript will require additional experiments as outlined below, before publication can be considered.

Main comments:

1. The authors restrict their mutagenesis analysis (Figs. 4, S4, S5) to two genetic variants in Znf6 and Znf7 that are likely causative for Okhiro syndrome. While I agree that these variants are interesting because of disease association, it would be valuable to do a more comprehensive mutational analysis to test the predictions from their structural studies.

For e.g., the authors state that "two small aliphatic residues, I897 and V925 provide hydrophobic surfaces that interact with methyl groups at C5 on thymine that point into the major groove. These hydrophobic surfaces likely provide some specificity of thymine bases within SALL4-associated sequences". Selex could be used to test this possibility by mutating these residues, as well as other residues guided by the analysis of the protein structure in complex with DNA.

2. The authors conclude that SALL4 has an expanded specificity for a range of AT rich sequences. Because there are many AT rich sequences throughout the genome, it is not clear what directs binding to actual target genes. In contrast, the study by Kong et al used a protein binding microarray to identify "an AT rich sequence with little degeneracy: AA[A/T]TAT[T/G][A/G][T/A], in which the WTATB in the center of the motif represents the core sequence." Kong et al also derived an in vivo consensus motif that agrees with the in vitro determined motif by profiling SALL4 binding across the genome using CUT&RUN. Does the Kong core or more extended site represent a preferred higher affinity site? This could be tested by EMSA and isothermal titration calorimetry. If this is the preferred binding site how would this inform their structural data?

3. Fig. 4b shows EMSAs for wild type and mutant SALL4 fragments of ZnF 6 & 7. If I am interpreting the figure correctly, the bottom of the gel shows free probe next to the label "0.5 μ M DNA"; the top of the wild type gel shows the shifted SALL4/DNA complex next to the label "SALL4/DNA". It looks like less free DNA probe is depleted in the mutants compared with the wild type protein, consistent with reduced binding. However, it is not clear why there is no visible Sall4/DNA complex at all for the mutants, at least at the higher protein concentrations.

To rigorously conclude that these disease associated mutations in Znf 6 & 7 have reduced binding, the authors need to use EMSA and/or isothermal titration calorimetry to measure the apparent Kd comparing wild type and mutant proteins. This would also determine if the reduced binding affinity for R900W and G921D really is as strong as the effect of H898R, which is expected to disrupt the fold of the protein. It would also be reassuring if this result was validated by EMSA in cells with full length proteins- for e.g. using the SALL4 constructs and cells used for the IF experiments (Fig. 4c). Testing for altered binding of mutant proteins at some native target genes (based on their ChIP-seq and RNA seq in Pantier et al Mol Cell 2021) would also be valuable to support the conclusion that the proposed Okhiro mutants R900W and G921D show reduced binding at functionally relevant sites.

4. Western blots to show comparable expression of wild type and mutant proteins should be included to confirm that significantly lower levels of protein expression do not at account for the apparent different cellular co-localization (overlap or not with DAPI) in

Reviewer #2 (Comments to the Authors (Required)):

Summary

SALL4 is a transcription factor responsible for maintaining vertebrate embryonic stem cell identity. It does this by binding a wide range of AT-rich DNA sequences via its zinc finger domains. Previous work has shown that the c-terminal ZFC4 domain of SALL4 plays an important role, however, how ZFC4 recognises these motifs was not fully understood. Watson et al present a crystal structure of ZFC4 bound to AT-rich DNA. This allows the characterization of SALL4 binding to DNA, finding that ZFC4 uses small hydrophobic and polar side chains to provide recognition. Additionally, Watson et al explore the effect of disease-causing mutations in SALL4, suggesting that these lead to reduced DNA binding but do not affect its preference for AT-rich sequences. Overall, this manuscript is well written.

Main points

Authors find that SALL4 recognises AT-rich DNA sequences through hydrophobic and polar side chains. The structural data in this paper strongly support this claim allowing identification of the binding residues between ZFC4 and the major groove of the DNA. However, some changes could be made to strengthen this argument.

- Authors state that the ZFC4 sequence is conserved in SALL1/3 and 4 and therefore should have an identical DNA binding specificity. If authors could show this by replicating EMSA or cell data with SALL1/3 this would greatly improve the impact of this paper by highlighting the general mechanism of AT-rich recognition. It should be noted that although these experiments (3 months) would improve the manuscript, I do not believe they are necessary for publication.
- Authors elegantly use the gnomAD database to highlight the importance of the ZFC4 region in SALL4 function. Could the authors extend this analysis to SALL1/3 to further support this. For example, is the VdVp ratio/mutations the same in those proteins?
- Given the structure could the authors discuss why GC rich DNA would not fit?
- Would it be possible to generate a mutant which loses its preference for AT rich DNA such as I897/V925?

Authors find that disease causing mutations in ZFC4 lead to reduced DNA binding not loss of AT-rich affinity. Watson et al nicely combine in vitro and cellular data to support this claim.

- However, the EMSA data shown appears to show complete loss of binding for DNA compared to HT-SELEX. Could the authors explain this discrepancy or use another DNA probe for binding.

Minor points

- 1) This study focuses on ZFC4. However, SALL4 contains 2 other ZFCs. Can the authors clarify what role they play in SALL4 function and whether they contribute to DNA recognition in results paragraph 1?
- 2) Please could the authors add the other 9 mutations mentioned in results paragraph 2 onto fig 1a.
- 3) "Kratky analysis of these data indicate that ZFC4 is highly dynamic" Can authors explain Kratky analysis and add a reference.
- 4) Figure 2 would benefit from a zoom out with the entire structure to help orientate the reader.
- 5) Green and blue dotted lines need to be clearer in figure 2b.
- 6) I believe the manuscript would benefit from combining figures 3 and 4. Both figures deal with how mutations effect binding to DNA and would make it easier for the reader if these were together.
- 7) In figure 4 add a wider field of view of the cells. This will allow the reader to gain an appreciation of how the mutations effect multiple cells.
- 8) Please add the analysis done in Supplemental figure 4c to figure 4c. This is crucial for the reader to interpret the data.
- 9) Please clarify the n in experiment supplemental figure 4c. Add number of experiments and number of cells analysed.
- 10) Fig 5a is a little unclear. Can authors add numbers to the figure to indicate the order of the process?
- 11) In the methods, can the authors add the objective used on the Zeiss microscope to take the immunofluorescence images.

Reviewer #3 (Comments to the Authors (Required)):

This manuscript by Watson and colleagues describes the structure, cellular localisation and DNA binding properties of the c-terminal Zn-finger domain (Zn4) of SALL4, a protein that is involved in vertebrate development. A crystal structure of a Zn4 bound to AT-rich DNA reveals the molecular details of their interface, and is complemented by biochemical and biophysical analysis of their interaction. Mutations of Zn4 in SALL4 that have been observed in Okinhiro syndrome were characterised and found to disrupt its DNA binding and cellular localisation.

I can't comment on the value of this work for the field at large, given that i do not work on SALL4 or vertebrate development and can only judge it technically. In that regard, this is a robust piece of work that spans structural biology, biochemistry, biophysics, cell biology and genetics. It was easy to read, their data is sound and the interpretation is commensurate with their results. I have no issues/conflicts with any of their conclusions and so recommend this for publication. My only minor comments/queries are the following:

- What are the little grey filled circles on the DNA schematic shown in figure 2b? They are not described in the figure legend.
- Could the specificity of Zn4 for AT-rich DNA might comes from DNA shape/deformability sensing, rather than direct readout of

base edges?

Reviewer #1 (Comments to the Authors (Required)):

SALL4 is a multi-zinc finger (ZnF) factor that is mutated in Okhiro syndrome; its expression is reactivated in many aggressive cancers where it has been considered as a candidate therapeutic target. However, its mode of interaction with DNA is not well understood. Two recent studies, one from this group of authors (Pantier et al Mol Cell 2021) and another investigating liver cancer (Kong et al Cell Reports 2020) found that the C terminal pair of ZnFs 6 & 7 in SALL4 bind AT rich sequences and are critical for the function of the protein.

In this manuscript, the authors describe the structure of the C-terminal ZnF cluster 4 of Sall4 in contact with an AT rich DNA motif. The paper is a focused study that provides valuable structural information about this important transcriptional regulator, though the results agree with prior analyses of how ZnFs interact with DNA, somewhat reducing the overall significance of this work for the field. In addition, I find that the functional analysis of the SALL4 mutants and the studies investigating the specificity/affinity of the SALL4 binding site are superficial. In my opinion the manuscript will require additional experiments as outlined below, before publication can be considered.

Main comments:

1. The authors restrict their mutagenesis analysis (Figs. 4, S4, S5) to two genetic variants in *Znf6* and *Znf7* that are likely causative for Okhiro syndrome. While I agree that these variants are interesting because of disease association, it would be valuable to do a more comprehensive mutational analysis to test the predictions from their structural studies.

For e.g., the authors state that "two small aliphatic residues, I897 and V925 provide hydrophobic surfaces that interact with methyl groups at C5 on thymine that point into the major groove. These hydrophobic surfaces likely provide some specificity of thymine bases within SALL4-associated sequences". Selex could be used to test this possibility by mutating these residues, as well as other residues guided by the analysis of the protein structure in complex with DNA.

Three additional mutations I897S, V925S and a dual I897S-V925S have been tested in both EMSA assays and in cell based assays (Fig. 3). These point mutations have more subtle effects than the disease mutations. As predicted, mutation of I897S (an aliphatic to polar change) reduces the affinity of SALL4 for DNA and shows loss of localisation to heterochromatic foci. We have not carried out SELEX experiments as it would not have been possible to do this extended analysis in the time given. We provide both *in vitro* and cell-based data to show the importance of this residue for DNA binding. The double mutation largely reflects the behaviour of the I897S mutation.

Interestingly, the V925S mutation showed similar DNA binding activity to the wild type *protein in vitro* and retained localisation to AT-rich sequences in cells. While this result is somewhat unexpected, it fits with our observation that the majority of residues that make up the DNA major groove recognition surface of SALL4 are small polar residues that support recognition of AT-rich sequences and are at the core of SALL4's ability to recognise a variety of different sequences. Note that the I897S/V925S double mutant behaves much as the I897S mutation. This shows that I897 has the greater impact on specificity and affinity. In previous work, we demonstrated that a T919D/N922A double mutant substantially decreased DNA binding. All of these results are consistent with mutagenic studies in Ru et al 2022 where alanine mutations of I897 (I887 in Human), N922 (N912 in Human) and T919 (T909 in Human) disrupt DNA binding. The first paragraph of the discussion has been expanded to address this point.

2. The authors conclude that SALL4 has an expanded specificity for a range of AT rich sequences. Because there are many AT rich sequences throughout the genome, it is not clear what directs binding to actual target genes. In contrast, the study by Kong et al used a protein binding microarray to identify "an AT rich sequence with little degeneracy: AA[A/T]TAT[T/G][A/G][T/A], in which the WTATB in the center of the motif represents the core sequence." Kong et al also derived an *in vivo* consensus motif that agrees with the *in vitro* determined motif by profiling SALL4 binding across the genome using CUT&RUN. Does the Kong core or more extended site represent a preferred higher affinity site? This could be tested by EMSA and isothermal titration calorimetry. If this is the preferred binding site how would this inform their structural data?

We identified 'ATATT' as the preferred DNA motif, which agrees with the '[A/T]TAT[T/G]' core motif identified by Kong et al., and the DNA sequence that we used for EMSA contains this core motif. Similarly, a recent study (Ru et al 2022 JBC, PMID: 36257403) showed that SALL4 ZFC4 binds to a core 'TATT' sequence, which is also in agreement with our observations.

While 'ATATT' is the best motif, our HT-SELEX data (both in Pantier et al, 2021 and in the current study) indicates that SALL4 ZFC4 is also able to bind to a wide range of other short AT-rich sequences to similar extents (Fig. S6a). We have added a sentence to the results (page 5 paragraph 2) to indicate how our data fit with prior observations by ourselves and others.

3. Fig. 4b shows EMSAs for wild type and mutant SALL4 fragments of ZnF 6 & 7. If I am interpreting the figure correctly, the bottom of the gel shows free probe next to the label "0.5 μM DNA"; the top of the wild type gel shows the shifted SALL4/DNA complex next to the label "SALL4/DNA". It looks like less free DNA probe is depleted in the mutants compared with the wild type protein, consistent with reduced binding.

However, it is not clear why there is no visible Sall4/DNA complex at all for the mutants, at least at the higher protein concentrations.

Loss of free probe as DNA is titrated with protein indicates a binding event even when there is no clearly shifted band on the gel. The smearing of the signal seen on the EMSAs for the mutant proteins is a common observation for proteins that interact weakly with nucleic acids. In these cases, it is likely that the reduced affinity of the mutant proteins for the DNA leads to a higher off rate of dissociation. In consequence, DNA-protein complexes are not stable while separating on the EMSA gel and so a continuum of signal (smear), rather than a distinct band, is observed (see PMID: 17703195, Table 3 for a further explanation). We have redone the EMSA assays including poly dI/dC to act as a cold competitor for non-specific binding. This has reduced the smearing on the gels.

To rigorously conclude that these disease associated mutations in Znf 6 & 7 have reduced binding, the authors need to use EMSA and/or isothermal titration calorimetry to measure the apparent Kd comparing wild type and mutant proteins. This would also determine if the reduced binding affinity for R900W and G921D really is as strong as the effect of H898R, which is expected to disrupt the fold of the protein. It would also be reassuring if this result was validated by EMSA in cells with full length proteins-for e.g. using the SALL4 constructs and cells used for the IF experiments (Fig. 4c). Testing for altered binding of mutant proteins at some native target genes (based on their ChIP-seq and RNA seq in Pantier et al Mol Cell 2021) would also be valuable to support the conclusion that the proposed Ohikiri mutants R900W and G921D show reduced binding at functionally relevant sites.

We have quantified all EMSA assays to estimate apparent binding constants for interactions (Figure 3c, 4e, Table S4). Furthermore, the level of enrichment measured by HT-SELEX (Figure 5b) is proportional to the affinity of a transcription factor for a given DNA motif (Jolma et al, 2010). Therefore, we can say confidently that G921D/R900W mutants have a significantly lower affinity for AT-rich motifs compared to wild-type ZFC4.

While additional characterisation could be useful, we believe that our experimental data *in vitro* (EMSA/HT-SELEX) and in cells (immunofluorescence) demonstrate that R900W and G921D mutations result in dramatically decreased DNA binding. We would anticipate that these mutations would show an almost complete loss of SALL4 binding genome-wide, similar to the “ZFC4mut” (i.e. T919D,N922A double mutation) ChIP-seq data characterised in Pantier et al, 2021, Figure 2.

As the majority of the SALL4 protein is natively unstructured, carrying out *in vitro* experiments on full length protein are challenging. However, all cell-based experiments are done with full length protein and are consistent with the *in vitro* work on shorter constructs.

4. Western blots to show comparable expression of wild type and mutant proteins should be included to confirm that significantly lower levels of protein expression do not account for the apparent different cellular co-localization (overlap or not with DAPI) in Fig. 4c.

This is an important technical point, and we thank the reviewer for raising this concern. As cells are transfected with variable efficiencies, western blot analysis (‘bulk’ levels of SALL4) is not appropriate to compare expression levels between WT and mutant constructs. To circumvent this problem, we have re-analysed immunostaining data and show that the level of expression of SALL4 in single cells does not affect nuclear pattern (Figure S5c of high/low expression of WT and mutant SALL4 at the same brightness/contrast ratio).

Reviewer #2 (Comments to the Authors (Required)):

Summary

SALL4 is a transcription factor responsible for maintaining vertebrate embryonic stem cell identity. It does this by binding a wide range of AT-rich DNA sequences via its zinc finger domains. Previous work has shown that the c-terminal ZFC4 domain of SALL4 plays an important role, however, how ZFC4 recognises these motifs was not fully understood. Watson et al present a crystal structure of ZFC4 bound to AT-rich DNA. This allows the characterization of SALL4 binding to DNA, finding that ZFC4 uses small hydrophobic and polar side chains to provide recognition. Additionally, Watson et al explore the effect of disease-causing mutations in SALL4, suggesting that these lead to reduced DNA binding but do not affect its preference for

AT-rich sequences. Overall, this manuscript is well written.

Main points

Authors find that SALL4 recognises AT-rich DNA sequences through hydrophobic and polar side chains. The structural data in this paper strongly support this claim allowing identification of the binding residues between ZFC4 and the major groove of the DNA. However, some changes could be made to strengthen this argument.

- Authors state that the ZFC4 sequence is conserved in SALL1/3 and 4 and therefore should have an identical DNA binding specificity. If authors could show this by replicating EMSA or cell data with SALL1/3 this would greatly improve the impact of this paper by highlighting the general mechanism of AT-rich recognition. It should be noted that although these experiments (3 months) would improve the manuscript, I do not believe they are necessary for publication.

Unfortunately, in the timescale given for revisions, it was not possible to generate additional DNA binding data for SALL1 and SALL3. It should be noted that were we to generate equivalent constructs for *in vitro* work, these would be almost identical in sequence to SALL4, with none of the residues interacting with DNA different between the expressed proteins (see protein alignment in Fig. S2c). Understanding whether residues outside of the DNA binding site can influence binding affinity or specificity is interesting but would be outside of the scope of this study.

There is substantial evidence that SALL1 and SALL3 bind to AT-rich DNA. For example Yamashita et al (2007, doi 10.1111/j.1365-2443.2007.01042.x) showed that SALL1 enriches for AT-rich sequences *in vitro* and binds to DAPI foci in mouse cells. Our previous mass spectrometry experiment to search for AT-rich binders isolated both SALL1 and SALL3 in addition to SALL4 (Pantier et al 2021 doi 10.1016/j.molcel.2020.11.046). We have added some additional background on SALL1 and SALL3 to the introduction and highlighted how the sequence similarities in ZFC4 are relevant to SALL1 and SALL3 biology in the discussion.

- Authors elegantly use the gnomAD database to highlight the importance of the ZFC4 region in SALL4 function. Could the authors extend this analysis to SALL1/3 to further support this. For example, is the VdVp ratio/mutations the same in those proteins?

We have extended the analysis and provided VdVp ratios for SALL1 and SALL3 (Figure S2d). There are some common trends between the three proteins but there are also some differences. Note that while SALL4 and SALL1 show similar expression profiles and are associated with genetic diseases that have overlapping presentations (Townes-Brock syndrome and Okhiro syndrome), SALL3 proteins have an apparently different expression profile and affect different organ systems (Parrish et al 2004, PMID: 15282310). It is therefore expected that SALL3 would have a different pattern of tolerance for missense mutations compared to SALL1 and SALL4.

- Given the structure could the authors discuss why GC rich DNA would not fit?
We have added a point to the second paragraph of the discussion to address this.

- Would it be possible to generate a mutant which loses its preference for AT rich DNA such as I897/V925?
These mutants have been generated and analysed (see reviewer 1 comments above). These data are in Figure 3.

Authors find that disease causing mutations in ZFC4 lead to reduced DNA binding not loss of AT-rich affinity. Watson et al nicely combine *in vitro* and cellular data to support this claim.

- However, the EMSA data shown appears to show complete loss of binding for DNA compared to HT-SELEX. Could the authors explain this discrepancy or use another DNA probe for binding.

Quantification of EMSA data suggest that DNA binding for R900W is substantially reduced, while the G921D mutation showed no apparent binding in this assay. As noted above to Reviewer 1, while there is no higher band on the EMSAs representing the DNA/protein complex with the SALL4 disease mutations, it is unlikely that there is a complete loss of binding to DNA when the “free” DNA band disappears as the protein is titrated in e.g. for mutation R900W. As noted above for reviewer 1, HT-SELEX enrichment is dependent on the binding constant (Jolma et al, 2010). Consequently, the HT-SELEX is still reporting on DNA binding albeit for very low affinity binders.

Minor points

1) This study focuses on ZFC4. However, SALL4 contains 2 other ZFCs. Can the authors clarify what role they play in SALL4 function and whether they contribute to DNA recognition in results paragraph 1?

We have added further background to the function of ZFC1 and ZFC2 in the introduction (paragraph 2). We have focused on ZFC4 as this zinc finger cluster appears to play a defining role in maintaining identity of ESCs through its recognition of AT-rich sequences. Ru et al JBC 2022 have shown that ZFC1 binds DNA and so may contribute to SALL4 function. However, we previously showed that constructs lacking ZFC1 and ZFC2 show similar phenotypic outcomes to wild type protein in ESCs (Pantier et al 2021), so it is unlikely that ZFC1 and ZFC2 make major contributions to DNA binding.

2) Please could the authors add the other 9 mutations mentioned in results paragraph 2 onto fig 1a.

We have redone the ClinVar search and added an updated set of variants of uncertain significance (there are now 12 that do not have an equivalent mutation in gnomAD). These are now mapped in Figure 1a.

3) "Kratky analysis of these data indicate that ZFC4 is highly dynamic" Can authors explain Kratky analysis and add a reference.

Additional explanation of the Kratky analysis and references have been added to the text (Page 4).

4) Figure 2 would benefit from a zoom out with the entire structure to help orientate the reader.

A zoomed out overview has been added to Fig. 2c.

5) Green and blue dotted lines need to be clearer in figure 2b.

Thicker lines have been added to Fig. 2b.

6) I believe the manuscript would benefit from combining figures 3 and 4. Both figures deal with how mutations effect binding to DNA and would make it easier for the reader if these were together.

The original Figures 3 and 4 have been combined into a new figure 4.

7) In figure 4 add a wider field of view of the cells. This will allow the reader to gain an appreciation of how the mutations effect multiple cells.

We used the 100x objective to obtain high resolution pictures showing nuclear organisation, so a wider view is not possible. Additional examples of representative cells are presented in Fig. S5c, and all raw and processed microscopy data will be provided on a shared server (<https://datashare.ed.ac.uk/handle/10283/4011>). Furthermore, all transfected cells were analysed and quantified in Fig. 4g (formerly Fig. S4c).

8) Please add the analysis done in Supplemental figure 4c to figure 4c. This is crucial for the reader to interpret the data.

This has been done, this is now panel Fig. 4g.

9) Please clarify the n in experiment supplemental figure 4c. Add number of experiments and number of cells analysed.

The number of cells analysed is indicated in the chart above each mutant (Fig. 4g). The legend now clarifies that each point on the chart is from an independent transfection.

10) Fig 5a is a little unclear. Can authors add numbers to the figure to indicate the order of the process?

This has been done.

11) In the methods, can the authors add the objective used on the Zeiss microscope to take the immunofluorescence images.

This has been added.

Reviewer #3 (Comments to the Authors (Required)):

This manuscript by Watson and colleagues describes the structure, cellular localisation and DNA binding properties of the c-terminal Zn-finger domain (Zn4) of SALL4, a protein that is involved in vertebrate development. A crystal structure of a Zn4 bound to AT-rich DNA reveals the molecular details of their interface, and is complemented by biochemical and biophysical analysis of their interaction. Mutations of Zn4 in SALL4 that have been observed in Okihiro syndrome were characterised and found to disrupt its DNA binding and cellular localisation.

I can't comment on the value of this work for the field at large, given that I do not work on SALL4 or vertebrate development and can only judge it technically. In that regard, this is a robust piece of work that spans structural biology, biochemistry, biophysics, cell biology and genetics. It was easy to read, their data is sound and the interpretation is commensurate with their results. I have no issues/conflicts with any of their conclusions and so recommend this for publication. My only minor comments/queries are the following:

- What are the little grey filled circles on the DNA schematic shown in figure 2b? They are not described in the figure legend.

These are the methyl groups of the T bases. This has been clarified in the figure legend.

- Could the specificity of Zn4 for AT-rich DNA might come from DNA shape/deformability sensing, rather than direct readout of base edges?

As shown in Fig. S1b, we observe some compression of the minor groove and opening of the major groove in the structure. It is likely that some level of deformability contributes to recognition of AT-rich sequences. However, given that there are many direct interactions with bases in the major groove, it seems likely that the selection of AT-rich sequences is primarily based on the interactions with small hydrophobic and polar amino acids.

December 16, 2022

RE: Life Science Alliance Manuscript #LSA-2022-01588-TR

Prof. Atlanta G Cook
University of Edinburgh
Wellcome Trust Centre for Cell Biology
Michael Swann Building, King's Buildings
Max Born Crescent
Edinburgh EH9 3BF
United Kingdom

Dear Dr. Cook,

Thank you for submitting your revised manuscript entitled "Structure of SALL4 zinc-finger domain reveals link between AT-rich DNA binding and Okhiro syndrome". We would be happy to publish your paper in Life Science Alliance pending final revisions necessary to meet our formatting guidelines.

-please upload your supplementary figures as single files and add the supp. Figure legends to the main manuscript text
-please upload your table files as editable doc or excel files or make sure that they are in the doc file of your main manuscript text

A. FINAL FILES:

B. MANUSCRIPT ORGANIZATION AND FORMATTING:

Sincerely,

Reviewer #1 (Comments to the Authors (Required)):

The authors' revised manuscript has addressed all my concerns.

Reviewer #2 (Comments to the Authors (Required)):

Watson et al have dealt with all my comments. The paper is well written and the data strongly supports their claims. I recommend it for publication.

December 20, 2022

RE: Life Science Alliance Manuscript #LSA-2022-01588-TRR

Prof. Atlanta G Cook
University of Edinburgh
Wellcome Trust Centre for Cell Biology
Michael Swann Building, King's Buildings
Max Born Crescent
Edinburgh EH9 3BF
United Kingdom

Dear Dr. Cook,

Thank you for submitting your Research Article entitled "Structure of SALL4 zinc-finger domain reveals link between AT-rich DNA binding and Okihiro syndrome". It is a pleasure to let you know that your manuscript is now accepted for publication in Life Science Alliance. Congratulations on this interesting work.

DISTRIBUTION OF MATERIALS:

Again, congratulations on a very nice paper. I hope you found the review process to be constructive and are pleased with how the manuscript was handled editorially. We look forward to future exciting submissions from your lab.

Sincerely,
